# Research

evolution

fusion, fission, chromosome number, sex chromosome, karyotype, genome size

**Author for correspondence:**
Heath Blackmon
e-mail: blackmon@tamu.edu

# Lineage-specific patterns of chromosome evolution are the rule not the exception in Polyneoptera insects

Terrence Sylvester[1], Carl E. Hjelmen[1], Shawn J. Hanrahan[2], Paul A. Lenhart[2], J. Spencer Johnston[2] and Heath Blackmon[1]

[1]Department of Biology, and [2]Department of Entomology, Texas A&M University, College Station, TX 77843, USA

TS, 0000-0001-6683-0793; CEH, 0000-0003-3061-6458; PAL, 0000-0003-1365-1470; JSJ, 0000-0003-4792-2945; HB, 0000-0002-5433-4036

The structure of a genome can be described at its simplest by the number of chromosomes and the sex chromosome system it contains. Despite over a century of study, the evolution of genome structure on this scale remains recalcitrant to broad generalizations that can be applied across clades. To address this issue, we have assembled a dataset of 823 karyotypes from the insect group Polyneoptera. This group contains orders with a range of variations in chromosome number, and offer the opportunity to explore the possible causes of these differences. We have analysed these data using both phylogenetic and taxonomic approaches. Our analysis allows us to assess the importance of rates of evolution, phylogenetic history, sex chromosome systems, parthenogenesis and genome size on variation in chromosome number within clades. We find that fusions play a key role in the origin of new sex chromosomes, and that orders exhibit striking differences in rates of fusions, fissions and polyploidy. Our results suggest that the difficulty in finding consistent rules that govern evolution at this scale may be due to the presence of many interacting forces that can lead to variation among groups.

## 1. Introduction

Chromosome number is one of the fundamental characteristics of a genome. It is also the first information collected about most genomes. In fact, the first chromosome counts were recorded prior to the development of the chromosome theory of inheritance [1]. Despite this early start, consistent rules governing the evolution of chromosome number across large clades remain elusive.

Changes in chromosome number can happen due to several mechanisms. We use the term fusion and fission to describe a decrease or an increase of one in chromosome number, respectively. However, these terms are simplifications and may represent multiple processes at the molecular level. Reduction in chromosome number can happen through Robertsonian translocations with the loss of non-essential DNA [2] or happen through the fusion of two chromosomes at the telomeres followed by loss of one of the centromeres [3,4]. By contrast, increases in chromosome number can occur due to simple chromosome fission in the centromere region [5] or due to the duplication of an entire chromosome. Changes in chromosome number of more than one can also occur. Although rare in most animal groups, demiploidy describes an increase chromosome number by one-half. Demiploidy events can occur by the joining of haploid gamete with an unreduced diploid gamete [6]. Finally, whole-genome duplication can lead to a doubling of chromosome number [7].

These changes in chromosome number can have broad impacts on gene transcription, recombination rates and sex chromosome evolution. The presence

of an extra copy of a chromosome can lead to both increases or decreases in gene transcription [8–10]. It has long been recognized that chromosome number should positively correlate with genome-wide recombination rates [11]. The frequency of recombination events and the proper segregation of chromosomes into gametes is dependent on crossing over in meiosis. The lower limit of the number of crossing over events is controlled by the number of chromosome arms in most species and by the number of chromosomes in some species [12]. This relationship between chromosome number and recombination has been suggested as a source of indirect selection on chromosome number in Hymenoptera though, recent analysis suggest this may be only a weak force [13,14]. Changes in chromosome number may also impact the evolution and behaviour of sex chromosomes. For instance, if chromosomes are broken into smaller chromosomes while keeping all else equal (e.g. genome size), the average chromosome size should be negatively correlated with the number of chromosomes. This can have important impacts on the fate of sex chromosomes. A comparative study of Coleoptera has shown that species are more likely to lose the Y chromosome and transition from XY to XO if they have many small chromosomes rather than a few larger chromosomes [15].

In sexual species, it is often assumed that changes in chromosome number are underdominant–heterozygotes have reduced fitness [16]. Chromosomal heterozygosity occurs when the chromosome complement from the parents differs (eg. if one parent contributes a fused chromosome). Perhaps the most widely known example of this is hybridization between horses and donkeys where the offspring carries 32 chromosomes from the mother and 31 chromosomes from the father and is sterile [17]. However, in wild mice which are heterozygous for a single fusion between chromosomes 16 and 17, there is no significant reduction in fertility and thus no reduction in fitness [18]. A large number of crosses in lemurs (where the chromosome number ranged from 44 to 60) exhibit a full range of fitness effects in crosses with parents that have different chromosome numbers [19]. By contrast, one can hypothesize that changes in chromosome number might be less deleterious in asexual species since they do not have to pair with any other genome in the population. Consistent with this, many asexual species have considerable variation in chromosome number [20].

To better understand the dynamics of chromosome evolution, we have chosen to work with the insect clade Polyneoptera, which includes the orders Blattodea (roaches and termites), Dermaptera, Embiidina, Mantodea, Notoptera, Orthoptera (grasshoppers), Phasmatodea and Plecoptera. Polyneoptera show striking differences in chromosome number variation among orders. One of the central goals of this work was to determine if these differences are due to idiosyncratic rates and patterns of evolution in each order, or due simply to differences in phylogenetic history. Polyneoptera also have variation in sex chromosome systems and, include sexual and asexual lineages allowing us to explore interactions between these characteristics and chromosome number. We assembled a trait dataset of chromosome number, sex chromosome system (SCS), genome size and reproductive mode. We analysed these data in both a taxonomic and a phylogenetic framework to determine the impact of the sexual system on rates of chromosome number evolution, the source of transitions in SCSs, and identify differences in patterns of chromosome number evolution among orders.

# 2. Material and methods

## (a) Data collection and phylogenetic inference

We downloaded all available chromosome data for the insect clade Polyneoptera from the Tree of Sex database and supplemented this with extensive literature searches [21,22]. All these data are available at www.karyotype.org (electronic supplementary material, table S1). We also downloaded genome size data from the animal genome size database [23] and supplemented this dataset with 60 new genome size estimates to yield a final dataset of 185 genome size estimates (electronic supplementary material, table S2).

Using PyPHLAWD and Phylota, we assembled two sequence datasets one for all Polyneoptera and one focused on the order Phasmatodea (electronic supplementary material, table S3) [24,25]. Sequences were aligned and checked for quality using MAFFT v. 7 and Gblocks v. 0.91b, respectively [26,27]. Rogue taxa were identified (electronic supplementary material, figure S1) and removed with Mesquite v. 3.51 based on preliminary trees built with RAxML v 8.2.10 [28–30]. Our final alignment for Polyneoptera contained 232 taxonomic units with 73% missing data, while the final alignment for the insect order Phasmatodea contained 41 taxonomic units with 57% missing data. We conducted two independent BEAST v. 2.5 runs of 100 million generations to infer time-calibrated phylogenies under a relaxed lognormal clock, a birth–death model, GTR + G as the nucleotide substitution model, and priors on the age of seven nodes (electronic supplementary material, table S4) [31]. The initial 50% of each MCMC run was discarded based on evaluation with Tracer v. 1.7 [32], and 50 phylogenetic trees were randomly sampled from the post-burnin period of each run. The 100 sampled trees form the posterior distribution used for the analyses described below. This approach was repeated to build the phylogenies for Phasmatodea. For detailed methods of data collection and phylogenetic reconstruction, please see electronic supplementary material, methods.

## (b) Modelling chromosome number evolution

We used the R package chromePlus to estimate rates of chromosome number evolution [33]. We tested two versions of our model, a simple model with chromosome fission and fusion and a complex model which included fission, fusion and polyploidy. Although we use the terms fusion and fission for convenience, it should be noted that these are simply changes (decreases and increases, respectively) in the haploid number by one. Based on the likelihood ratio test results (discussed below), we used the complex model to estimate the rates of chromosome number evolution. To get reliable estimates for the rates of chromosome number evolution, we only analysed the four orders with at least 20 representatives.

To account for uncertainty in chromosome number (e.g. when there were reports of multiple values for a tip in our phylogeny), we randomly sampled among the possible values and repeated for each tree. To account for uncertainty in phylogeny, we ran an MCMC of 1000 generations for each of the 100 trees in the posterior distribution. Inspection of the parameter estimates revealed that our MCMC runs converged by 50 generations in most cases. We conservatively discarded the initial 25% as burnin and randomly sampled 100 states from the post-burnin portion of the run. This process yielded 10 000 point estimates that define the posterior distribution of the parameters in our model. We tested for differences in rates of chromosome number evolution by comparing the 95% credible interval of the posterior distribution for each parameter in our model. Rates were inferred with branch lengths transformed to make trees unit length. However, all rates reported have been back-transformed so they represent transition rates in units of millions

of years. As is customary for Markov models, the reported rates are lambda parameters for exponential distributions that describe the expected waiting time for a transition to occur. Since our tree has branch lengths in units of millions of years these are reported in units of per MY.

## (c) Genome size

We tested whether genome size as a proxy for repetitive content might explain variation in rates of chromosome number evolution. Expansions in genome size are largely due to repetitive content, especially transposable elements [34,35]. We reasoned that increased transposon activity could lead to a greater frequency of fusion and fission mutations and result in higher rates of chromosome number evolution in larger genomes. We also tested for a correlation between genome size and chromosome number, reasoning that recent whole-genome duplications should lead to an increase in both values.

We first tested whether genome size was predictive of chromosome number. For this analysis, we fit a linear model where genome size was the predictor variable and chromosome number was the response variable using all species with both chromosome number and genome size estimates ($n = 55$) [36]. This was repeated for a reduced dataset using a phylogenetically corrected linear model including only those taxa present on our phylogeny ($n = 23$) [37]. To test whether the genome size for a species predicted its rate of chromosome number evolution, we first calculated tip rates for all species on our phylogeny that also had genome size estimates ($n = 20$). This tip rate was calculated as the difference between the tip value and the most probable chromosome number of the immediate ancestor of a given tip (see below for ancestral state reconstructions of chromosome number) divided by the branch length between the ancestor and the tip. We use the mean of this value calculated across the posterior distribution of trees. We evaluated both an absolute tip rate and a directional tip rate (accounting for whether the change is an increase or decrease in chromosome number). We then fit both standard and phylogenetically corrected linear models where genome size predicted either the absolute rate or the directional rate [36,37].

## (d) Ancestral state reconstructions

We estimated the ancestral states of chromosome number and the sex chromosome system (SCS). We estimated the ancestral states of the chromosome number at the root of each order using chromEvol v. 2.0. [38,39]. We used a fixed parameter model which included chromosome gains, losses and whole-genome duplication—matching the model used in chromePlus. For each tree from our posterior distribution, we took the mean of each parameter estimate from the corresponding chromePlus analysis described above and supplied these to infer ancestral states in chromEvol. We combined the estimates from the analysis of all trees in the posterior.

The estimate of ancestral states for SCSs was done using the ARD model in the function ACE in the R package APE [40]. We classified multi-XY SCS as XY which resulted in two states (XO and XY). To estimate the number of transitions in SCSs, we created the same model and performed stochastic mappings in the R package phytools [41]. Data and all R code for analyses are provided in a GitHub repository: https://github.com/Tsylvester8/Polyneoptera.

# 3. Results

## (a) Evolution of sex chromosome systems

In our dataset, 23 genera (182 taxa) contain species with at least two types of SCSs (i.e. XO, XY, or multi-XY). In each of these genera, we compared the mean haploid autosome number for all species with a given SCS. By comparing these means within genera, we can determine if differences are consistent with fusions or fissions as a source of transitions among SCSs. Briefly, if transitions from XO to XY are generated by the fusion of an autosome to a sex chromosome, we would expect a lower mean autosome number for XY species. Likewise, if transitions from XO or XY to multi-XY are generated by the fusion of an autosome to a sex chromosome, we would expect a lower mean autosome number in multi-XY species. By contrast, if transitions from XY to multi-XY are generated by the fission of an existing sex chromosome, we would expect the mean autosome number to be unchanged in the multi-XY species. We find strong support for fusions as a source of transitions from XO to XY SCSs. Of the 17 genera with both XO and XY species, 94% (16/17) show a lower mean number of autosomes in XY species (table 1). However, we find support for both fusions and fissions leading to transitions from XY to multi-XY. Of the 10 genera evaluated 40% (4/10) have higher or unchanged mean number of autosomes in multi-XY species (suggestive of fissions). By contrast, 60% (6/10) of the genera have a lower mean autosome number in the multi-XY species (suggestive of fusions).

## (b) Ancestral states and rates of sex chromosome evolution

We find that the ancestral state for SCS in Polyneoptera clade was XO, with a probability of 90.3%. Similarly, the most probable ancestral state for each order was also XO, except for Isoptera and Dermaptera, where XY is more probable (figure 1). We find the credible intervals of the transition rates from XO to XY and XY to XO to be largely overlapping, with means of 0.00202 and 0.00200, respectively. However, transitions from XO to XY were more common (mean = 15.3), while transitions from XY to XO were relatively rare (mean = 6.7).

## (c) Rates of chromosome number evolution

Some Polyneoptera orders exhibit little variation in chromosome number while others are highly variable (figure 1; see electronic supplementary material for results of order level variances in chromosome number). This could be because some orders are evolving more quickly or it could be because their phylogenetic history has allowed for a greater period of divergence. To draw any rigorous conclusions, we must explicitly control for this history. We began by applying a base model that includes only fusions and fissions and compared this via a likelihood ratio test with a model that included polyploidy. This was repeated for each of the 100 trees from our posterior distribution for each order. All orders showed some support for the model including polyploidy and overall, 77.6% of our likelihood ratio tests supported the more complex model that included polyploidy. For this reason, all analyses were done with the model with fusions, fissions and polyploidy, allowing us to compare the same set of rates across all orders. In Blattodea (including Isoptera), we estimate a mean fusion rate of 0.128, a fission rate of 0.150 and a polyploidy rate of 0.003 (electronic supplementary material, table S5). By contrast, if we remove the subclade Isoptera from Blattodea we find that parameter estimates increase to 0.420, 0.385 and 0.004 for fusions, fissions and polyploidy, respectively. This is consistent with rate estimates for Isoptera in isolation, where we infer rates of

**Table 1.** Chromosome number and sex chromosome systems. Within each genus, we report the mean number of autosomes for all species having XO, XY, or complex sex chromosome systems. The last two columns states whether the given chromosome numbers support fusion or fission as an important process in transition between these sex chromosome systems. Negative (−) symbol indicates a distribution of chromosome number that does not support either mechanism.

| order | genus (samples) | XO | XY | multi-XY | XO to XY | XO or XY to multi-XY |
| --- | --- | --- | --- | --- | --- | --- |
| | | mean number of autosomes | | | | |
| Blattodea | *Cryptotermes* (6) | | 23 | 16.4 | | fusion |
| Dermaptera | *Forficula* (14) | | 11 | 10.8 | | fusion |
| | *Nala* (3) | | 17.5 | 17 | | fusion |
| | *Nesogaster* (2) | 10 | | 9 | | fusion |
| Mantodea | *Deiphobe* (2) | 9 | | 12 | | fission |
| Orthoptera | *Aleuas* (6) | 9 | 9.2 | | — | |
| | *Dichroplus* (35) | 10.74 | 8.71 | 9 | fusion | fission |
| | *Diponthus* (7) | 10.5 | 10 | | fusion | |
| | *Eurotettix* (5) | | 10 | 9 | | fusion |
| | *Gryllotalpa* (5) | 9.67 | 5 | | fusion | |
| | *Isophya* (25) | 15 | 14 | | fusion | |
| | *Leiotettix* (10) | 11 | 8 | 5.5 | fusion | fusion |
| | *Scotussa* (8) | 10.6 | 8.5 | 9 | fusion | fission |
| | *Scyllina* (3) | 11 | 10 | | fusion | |
| | *Tetrixocephalus* (5) | 11 | 10 | | fusion | |
| | *Xyleus* (9) | 11 | 10 | | fusion | |
| | *Zoniopoda* (6) | 11 | 10 | | fusion | |
| Phasmatodea | *Didymuria* (11) | 17.6 | 14.67 | | fusion | |
| | *Isagoras* (3) | 18 | 16 | | fusion | |
| | *Leptynia* (5) | 18.25 | 17 | | fusion | |
| | *Podacanthus* (3) | 17 | 13 | | fusion | |
| | *Prisopus* (2) | 24 | 13 | | fusion | |
| Plecoptera | *Perla* (7) | 9.5 | 4 | 11.75 | fusion | fission |

0.044, 0.063 and 0.003 for fusions, fissions and polyploidy, respectively. Mantodea rate estimates exhibited high uncertainty, overlapping rate estimates in most orders for most parameters (figure 2; electronic supplementary material, figure S2). Some of the lowest rates we estimated were in Orthoptera where the fusion rate was 0.003 and the fission rate was 0.024. However, we do find the polyploidy rate to be relatively high with a mean rate of 0.101.

### (d) Chromosome number evolution versus genome size

We performed genome size estimation of 60 Polyneoptera species from the orders Blattodea, Mantodea, Orthoptera and Phasmatodea (electronic supplementary material, table S2 and text). The largest of these was 18051.1 Mbp in *Hadrotettix trifasciatus* (Orthoptera), while the smallest was 2071 Mbp measured in *Thesprotia graminis* (Mantodea). These data, along with 125 publicly available genome size estimates, had an overlap of 55 species with our chromosome dataset and 23 species with our phylogenetic dataset.

We found a significant trend towards lower chromosome numbers with higher genome sizes ($p$-value = 0.01), but after correcting for phylogeny this was not significant ($p$-value = 0.75). This difference appears to be due to the largest genome sizes and some of the smallest chromosome numbers both

occurring in Orthoptera (electronic supplementary material, figure S3A). We found no significant relationship between directional tip rates and genome size (electronic supplementary material, figure S3B), or absolute tip rates (electronic supplementary material, figure S3C).

### (e) Asexuality and rates of chromosome number evolution

Our dataset contains 13 parthenogenetic Phasmatodea species. We tested whether the rates of chromosome number evolution are contingent on the reproductive mode (phylogeny and trait data is shown in electronic supplementary material, figure S5). We found that there is no significant difference in the rates of chromosome fusion and fission between sexually and asexually reproducing lineages (electronic supplementary material, figure S6A and S6B). However, we find that rates of polyploidy are significantly higher in asexually reproducing lineages than in sexually reproducing lineages (electronic supplementary material, figure S6C).

## 4. Discussion

The evolution of chromosome number across large clades and long time spans is fundamental to the diversity of

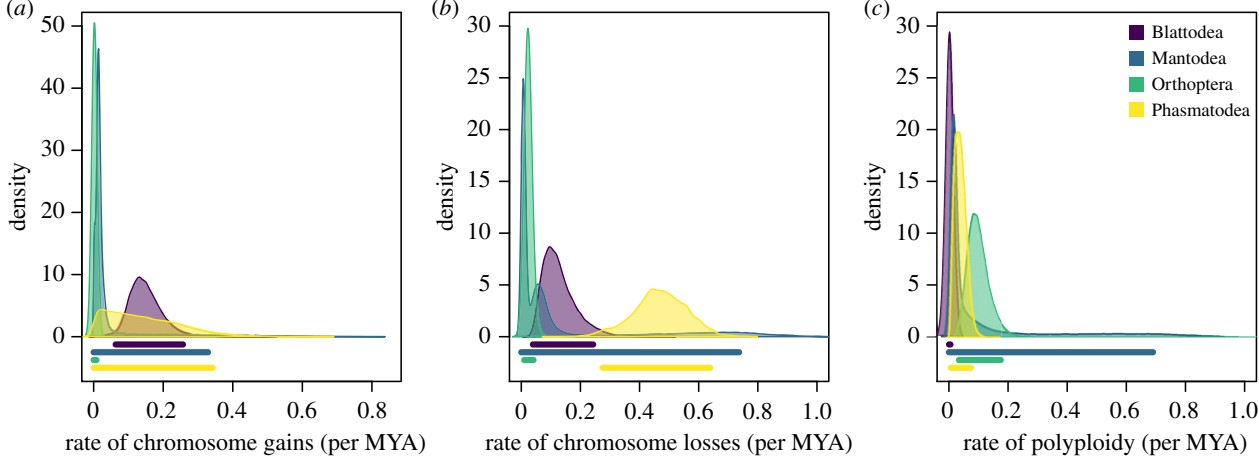

**Figure 1.** One of the 100 trees from the posterior distribution with chromosome number and sex chromosome system displayed. Bar heights represent the haploid chromosome number of each taxa. Tips are labelled according to the sex chromosome system/reproductive mode of the taxa. Tips that are marked as being XY includes species with XY and multi-XY sex chromosome systems. The pie charts at the roots of each order and at the root of the tree represent the probability of that node being either XO or XY, averaged across the posterior distribution of 100 phylogenies (in this analysis, we discarded the tips that are parthenogenetic and that did not have data for SCS). The rings represent the 25 and 50 chromosome number margin. (Online version in colour.)

**Figure 2.** Rates of chromosome (*a*) fission, (*b*) fusion and (*c*) polyploidy of four orders in the insect clade Polyneoptera. The bars below each distribution indicates the 95% HPD interval. Orders are indicated by the fill colour. (Online version in colour.)

royalsocietypublishing.org/journal/rspb　　Proc. R. Soc. B 287: 20201388

genomes we observe across the tree of life. Despite this, we are only beginning to understand how chromosome number evolves. In this study, we have focused on the dynamics of chromosome number evolution in Polyneoptera.

## (a) Sex chromosomes and chromosomes number evolution

The transition between SCSs from XO to XY can occur through the fusion of the X chromosome and an autosome or by sex chromosome turnover with fixation of the ancestral X as an autosome [16,42,43]. Transition via fusion will lead to a reduction in the total number of autosomes, while turnover should lead to no change in the total number of autosomes. Our data show a clear pattern of reduced autosome number in taxa with likely XO to XY transitions supporting fusions as a dominant pathway for this change (table 1). Transition from XY to multi-XY can occur through the fusion of a sex chromosome and an autosome or by sex chromosome fission. Transition via fusion will lead to a reduction in the mean number of autosomes while fissions will lead to no change in the number of autosomes. Our data show a mixed pattern with multi-XY species having both increased and decreased autosome number (table 1). We interpret this as evidence for both fusions and fissions as an important source of multi-XY systems. Combining our results for XO to XY and XY to multi-XY transitions, we find that fusions are a dominant route for changes in SCSs a pattern consistent with common sexual antagonistic variation on autosomes leading to selection for fusions [42,44]. The recent development of a null model for fusions that facilitates testing for an excess of sex chromosome autosome fusions is an obvious next step that could be applied in groups like Orthoptera and Phasmatodea [45].

Our genus-level analysis of autosome number and SCS is limited to only those genera with more than one SCS and thus omits much of our data. When autosome number among orders is parsed by SCS, differences in the mean autosome number suggest that in some groups the origin of transitions may differ from our genus-level analysis. For instance, the mean autosome number of all XY species in both Blattodea and Dermaptera is higher than the mean autosome number of XO species (electronic supplementary material, figure S7). This pattern is not expected if fusions are the primary source of transitions from XO to XY (table 1). However, without modelling both SCS and chromosome number jointly on a phylogeny these patterns are at best difficult to interpret.

Our assembled data can also help us understand the fate of the Y chromosomes. It has been suggested that Y chromosomes may be destined to decay and loss given the inevitability of mutation accumulation and reductions in the area that undergoes recombination and allows for segregation [46,47]. Alternatively, they may be retained through cycles of rejuvenation or even transitions into alternative forms of meiosis [15,48]. In our inference of SCS evolution, the rate of Y chromosome gains and Y chromosome losses are both approximately 0.002 (electronic supplementary material, table S6). However, we find that Y chromosome gains are more common with a mean of 15.3 across our entire phylogeny while losses are relatively rare with a mean of 6.7 across the entire phylogeny. This pattern is intuitive when we consider that the ancestor of this group was likely XO and thus there has been relatively little time for the gain of the Y chromosome to then be followed by its decay and loss.

## (b) Constraints on chromosome number evolution

In many clades, chromosome number is likely to change primarily through fusions and fissions of existing chromosomes [33,49]. However, chromosome number could also change due to aneuploidy or whole-genome duplication events that fix in a population, creating duplicate copies of one or more chromosomes. In fact, a recent analysis of 28 transcriptomes from Polyneoptera species revealed evidence for at least four independent whole-genome duplication events and two independent partial genome duplication events [50]. Parsing the relative contribution of fissions and aneuploidy to increases in chromosome number is not possible with our dataset, but could be tested with cross-species chromosome painting via fluorescence in situ hybridization [51]. The converse chromosome number decrease due to aneuploidy is likely to be exceedingly rare since all genes on the chromosome would have to be dispensable. However, these processes (fusion, fission, whole-genome duplication and aneuploidy) could all lead to sterile offspring if two populations (one with the chromosome duplication and one without) hybridize, since the heterozygous offspring may have difficulty segregating unmatched chromosomes during meiosis or gametes may carry an incomplete set of genes [52]. In both sexual and asexual species, chromosome increase due to aneuploidy may be rare due to the impact of aneuploidy on dosage which may lead to stoichiometric imbalances in gene networks. However, asexually reproducing species should be immune to the problem of proper segregation since they cannot outcross. For these reasons, we expected to see a higher rate of chromosome number increase and decrease in asexual species. Our Phasmatodea dataset has a mean of 9.3 transitions from sexual to asexual reproduction and offers a chance to test this hypothesis (electronic supplementary material, table S7). To our surprise, our analysis illustrates that rates of chromosome increase and decrease are equal in sexual and asexual Phasmatodea (electronic supplementary material, figure S6). We interpret this as evidence that the constraints on chromosome number change via fusions, fissions and aneuploidy are largely similar in sexual and asexual Phasmatodea. The most parsimonious explanation seems to be that the changes observed are largely neutral and that individuals that are heterozygous for chromosomal rearrangements do not typically have difficulty properly segregating chromosomes during meiosis. By contrast, the constraints on polyploidy appear to be lifted in asexual lineages, suggesting these changes may be deleterious in sexual species but neutral or nearly neutral in asexual species.

## (c) Variation in rates of chromosome number evolution

Most studies of chromosome number evolution have been done on small clades in isolation [53–55]. This creates a challenge in understanding variation in rates of chromosome number evolution across the tree of life since rates are fundamentally influenced by the time constraints and branch lengths inferred in a study (but see: [33,56]). By inferring rates in four orders all using a common tree, we are able to make a more valid comparison among clades and determine whether some groups are evolving at significantly different rates. We found many examples of significantly different rates of chromosome number evolution among orders. Blattodea have a higher rate of fissions than Orthoptera (figure 2a; electronic supplementary material, table S5). Blattodea (excluding Isoptera) have a higher

rate of fusions than Isoptera and Orthoptera (figure 2*b*; electronic supplementary material, table S5). Fusions are also higher in Phasmatodea than Blattodea, Isoptera and Orthoptera (electronic supplementary material, table S5). Polyploidy is higher in Orthoptera than Blattodea and, Phasmatodea is higher than Blattodea (figure 2*c*; electronic supplementary material, table S5). In line with existing evidence [21,57], polyploidy is higher in asexual than sexual species (electronic supplementary material, figure S6).

One possible explanation for variation in rates of chromosome number evolution is fundamental differences in the repeat content of the genome. For instance, large numbers or recent expansions of transposable elements may lead to more frequent chromosome breakage or other structural rearrangements that change chromosome number [58]. If transposable elements have expanded in the genomes of a clade, we might expect to see a signature of this in increased genome sizes [34]. However, we found no association between genome size and absolute rates of chromosome number evolution suggesting that repetitive content is not a driving force in the stability of large-scale genome structure across Polyneoptera. We also investigated the relationship between chromosome number and genome size. In particular, we hypothesized that if recent polyploidy events were present we should expect to find increases in both measures. Indeed a linear model with chromosome number as the response variable and genome size as the predictor variable is significant. However, this pattern is in the opposite direction from what we would expect due to polyploidy (smaller chromosome numbers are found in larger genomes). However, this pattern is driven largely by the low chromosome number and large genome size in Orthoptera, and once corrected for phylogeny, we find no significant relationship between these variables. We interpret this result as evidence that our dataset lacks any very recent polyploidization. An additional expectation for recent polyploids would be that they would exhibit a large positive tip rate (large increases in chromosome number since the most recent common ancestor) and a large genome size. However, we do not find any significant relationship between tip rates and genome size with or without correction for phylogeny (electronic supplementary material, figure S3). Moving forward, the recent development of multiple probabilistic models of chromosome number evolution that allow for associations with speciation or binary characters offers a way forward to further tease apart the determinants of rates of chromosome number evolution [33,59,60].

### (d) Reconciling a century of chromosome research
We find that many of our results confirm previous hypotheses on chromosome number evolution and SCS evolution in Polyneoptera. With respect to SCSs, Mantodea, Orthoptera and Phasmatodea had all previously been hypothesized to originate from XO ancestors and our results confirm these hypotheses [21,61,62]. Additionally, evidence that the X chromosome of the German cockroach, *Blattella germanica* (Blattodea), is homologous to the X chromosome in most Diptera is consistent with a shared XO SCS in the ancestor of all Polyneoptera [63]. However, other work has shown that the X chromosome in *Drosphila melanogaster* is not homologous to the Z in *Bombyx mori* or the X in *Tribolium castaneum*, or even the X in many flies, suggesting that insects may have frequent turnover in SCSs [43,64]. Even if early polyneopterans shared a common XO SCS, the high frequency of sex chromosome

autosome fusions and Y chromosome losses that we document suggest that the gene content of sex chromosomes of extant species is likely variable.

In other cases, the application of probabilistic models to our expanded dataset challenges previous hypotheses. For instance, it has been hypothesized that the most recent common ancestor of Blattodea (including Isoptera) was XY and the Y chromosome had been rapidly lost in Blattodea (excluding Isoptera), our results support an XO ancestor with a probability of 99.04% [65,66]. In Isoptera, it has been hypothesized that the ancestral SCS was XO, but our results suggest that ancestral SCS of Isoptera was in fact XY (with a probability of 89.44%) [62,65].

Even without model-based analyses, some authors have suggested that fusions or fissions were more important in some groups. For instance, in Isoptera, it has been previously hypothesized that fusions are more common than fissions [65,66]. Although we do not find a significant difference between the rates of fusion and rates of fission (electronic supplementary material, table S5), our ancestral state analysis for chromosome number finds that the average number of fusion events are significantly higher than number of fission events (33.32 fusion events and 21.94 fission events, *t*-test *p*-value less than 0.05). Our results depart most strongly from previous work in estimates of ancestral states (discussed in electronic supplementary material). For instance, in Mantodea, we inferred 8 and 7 as the most probable ancestral haploid number for the group while previous work predicted ancestral haploid number of 14 [61]. In Orthoptera, we inferred 6 as the most probable ancestral haploid number while previous work predicted ancestral haploid number of 12 [16]. Finally in Phasmatodea, we inferred 9 and 10 as the most probable ancestral haploid number while previous work predicted ancestral haploid number of 18 [67]. We note however, that our ancestral state estimates are dependent on the model applied. In our study, we used a model that allowed for a possibility of polyploidy in all orders. This has the impact of increasing the probability of low ancestral state estimates that may not be realistic if growing genome evidence finds reduced support for whole-genome duplication events in these orders. However, we note that our preliminary analyses showed that our finding of differences in rates among orders is not impacted by the inclusion or exclusion of polyploidy in our model.

## 5. Conclusion
Our analyses and synthesis point to numerous clades that should be targeted with future whole-genome sequencing projects. For instance, chromosome level genome assemblies of XY orthopterans would be a powerful tool to discover whether the same chromosome is repeatedly co-opted into new XY systems. Similarly, we have identified sister species in Dermaptera, like *Labidura riparia* and *Nala lividipes*, that, though closely related, have 12–14 and 34–40 chromosomes, respectively. Whole-genome sequencing and comparative genomics would allow us to better understand how these dramatic restructuring of karyotypes have occurred. Taken as a whole, our results illustrate that the striking differences in chromosome number variation among orders is due to differences in rates and patterns of chromosome number evolution within orders and not due simply to sampling or the age of different clades. With the exception of Mantodea, all

investigated orders had at least one transition rate that was different from one or more other orders. This has important implications for our understanding of the speciation processes. For instance, while many chromosomal speciation models [68,69] have been thought to be unimportant in recent times, it may be that in groups like Blattodea (excluding Isoptera), that exhibit high rates of chromosomal evolution, these models may explain an important source of extant diversity. Even if these older models do not represent a primary source of reproductive isolation, groups with higher rates of chromosome number evolution may be more likely to experience speciation. For instance, speciation may be facilitated under newer models of chromosomal speciation through sheltering of portions of the genome from admixture allowing incipient species to diverge and build-up genetic incompatibilities [70]. More broadly, depending on the importance of epistatic relationships, the reorganization of the genome through fusions and fissions may be important in determining the ability of organisms to adapt to novel environments [71].

Data accessibility. Code and data to perform all analyses reported are available via GitHub https://github.com/Tsylvester8/Polyneoptera and from the Dryad Digital Repository: https://doi.org/10.5061/dryad.tx95x69vq [72].

Authors' contributions. All authors contributed to the writing and revision of the manuscript.

Competing interests. We declare we have no competing interest.

Funding. This work was supported by National Institute of General Medical Sciences at the National Institutes of Health R35GM138098.

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
