## [Reviewer comments · Proceedings of the Royal Society B: Biological Sciences]

Review History

RSPB-2019-2544.R0 (Original submission)

Review form: Reviewer 1

Recommendation

Major revision is needed (please make suggestions in comments)

Scientific importance: Is the manuscript an original and important contribution to its field?

Acceptable

General interest: Is the paper of sufficient general interest?

Acceptable

Quality of the paper: Is the overall quality of the paper suitable?

Acceptable

Is the length of the paper justified?

No

Should the paper be seen by a specialist statistical reviewer?

Yes

Do you have any concerns about statistical analyses in this paper? If so, please specify them explicitly in your report.

Yes

It is a condition of publication that authors make their supporting data, code and materials available - either as supplementary material or hosted in an external repository. Please rate, if applicable, the supporting data on the following criteria.

Is it accessible?

N/A

Is it clear?

N/A

Is it adequate?

N/A

Do you have any ethical concerns with this paper?

Yes

Comments to the Author

Please see the attached file. (See Appendix A)

I noticed that the title is misleading and somewhat strange. I tried to look for "idiosyncratic" or synonyms in the main text but failed to identify anything corresponding with the title. This is yet another attempt to sell the data using a sexy and "cool" title. The title should be modified to better reflect the actual results. Alternatively, include some summary paragraph in Discussion justifying the current title.

Review form: Reviewer 2

Recommendation

Major revision is needed (please make suggestions in comments)

Scientific importance: Is the manuscript an original and important contribution to its field?

Excellent

General interest: Is the paper of sufficient general interest?

Excellent

Quality of the paper: Is the overall quality of the paper suitable?

Excellent

Is the length of the paper justified?

Yes

Should the paper be seen by a specialist statistical reviewer?

No

Do you have any concerns about statistical analyses in this paper? If so, please specify them explicitly in your report.

No

It is a condition of publication that authors make their supporting data, code and materials available - either as supplementary material or hosted in an external repository. Please rate, if applicable, the supporting data on the following criteria.

Is it accessible?

Yes

Is it clear?

Yes

Is it adequate?

Yes

Do you have any ethical concerns with this paper?

No

Comments to the Author

I enjoyed reading the present manuscript and I consider it a valuable contribution to our understanding of the role of genome organisation in evolution. The study is clearly a follow-up to the report published by Blackmon, Ross, and Bachtrog (2016). Therefore, the results should be compared in more detail to earlier conclusions in the discussion. Moreover, I came across several issues which need to be corrected or clarified.

line 80: "compliment" - should be "complement"?

lines 121-123: The source of sequences is not clear. Although processed alignments are available at <https://github.com/Tsylvester8/Polyneoptera>, accession numbers of original sequences of all species should be listed in Table S1. Or at least refer to the github repository.

lines 126-127 and 232: should not it be "species-level" and "genus-level" in all cases?

lines 143-144: How exactly was the inflexion point determined? I would say it is around the score of 3500.

lines 226-228 and 326-330: The authors state that 3/7 of the genera with both XY and multi-XY have a higher mean autosome number in multi-XY species, i.e. mean autosome number is increased, and this is interpreted as evidence for fissions. However, the authors reason that "if transitions from XY to multi-XY are generated by the fission of a sex chromosome we would expect no difference in the number of autosomes" for multi-XY species. This should be elaborated both in the results and discussion. I guess that the authors consider fissions responsible for increase of autosome number and therefore also for transition to multiple sex chromosome systems, although it does not fit the predicted pattern. But this is guilty by association. It was shown that autosomes and sex chromosomes evolve under different selection regimes, e.g. Ohno's law postulates that synteny of genes from the mammalian X chromosome is evolutionary more conserved compared to autosomes.

line 270: "0.063 and a polyploidy rate" to "0.063, and a polyploidy rate"

lines 311-312: "Our analyses have revealed that this clade originated from an XO ancestor." Support for a common origin, i.e. homeology, of XO sex chromosome system in Polyneoptera should be discussed (see DOI:10.1186/s12915-019-0721-x).

lines 319-320: Add references to the statement: "The transition between SCSs from XO to XY can occur through X chromosome autosome fusions or by sex chromosome turnover with fixation of the ancestral X as a new autosome."

lines 387-390: "... our analysis illustrates that rates of chromosome increase and decrease are equal in sexual and asexual Phasmatodea (Fig. 4). We interpret this as evidence that the constraints on chromosome number change via fusions, fissions, and aneuploidy are largely similar in these two groups." How could the authors determine whether the increase in chromosome number is caused by fissions or aneuploidy? Both result in increase in the haploid number by one. Could it be that there are similar rates of fissions and aneuploidies in sexuals and asexuals, respectively? Is there any hard evidence supporting increase in chromosome number via aneuploidy in Polyneoptera?

One of interesting results of present study is the role of presumably rare polyploidy in karyotype evolution in Polyneoptera. The more complex model including polyploidy was supported by likelihood ratio tests (line 264) and relatively high polyploidy rates were estimated for Mantodea, Phasmatodea, and Orthoptera. Do available genome sizes support polyploidy inferred from chromosome numbers? As it is not always the case in insects.

Page 13, Table 1.: "A - symbol indicates a distribution of chromosome number that is uninformative or does not support a given mechanism." I cannot see any As in the table.

Decision letter (RSPB-2019-2544.R0)

17-Dec-2019

Dear Mr Sylvester:

I am writing to inform you that your manuscript RSPB-2019-2544 entitled "Idiosyncratic patterns of chromosome evolution are the rule not the exception" has, in its current form, been rejected for publication in Proceedings B.

This action has been taken on the advice of referees, who have recommended that substantial revisions are necessary. With this in mind we would be happy to consider a resubmission, provided the comments of the referees are fully addressed. However please note that this is not a provisional acceptance.

Please note that this decision may (or may not) have taken into account confidential comments.

In your revision process, please take a second look at how open your science is; our policy is that all data involved with the study should be made openly accessible-- see: <https://royalsociety.org/journals/ethics-policies/data-sharing-mining/>
Insufficient sharing of data can delay or even cause rejection of a paper.

Sincerely,
Professor John Hutchinson, Editor
mailto: proceedingsb@royalsociety.org

Associate Editor

Board Member: 1

Comments to Author:

This manuscript addresses a fundamental question in chromosomal evolution: why does chromosome number vary so much between different lineages? The authors combine an extensive phylogenetic analysis with karyotypic information from the tree of sex database in the large insect group Polyneoptera to test some theories of what may drive these changes. The different hypotheses tested (sex chromosome evolution, asexuality, genome size) were not able to explain most of the variation that they observe.

This is in principle an excellent approach and model system to address a fascinating question. However, it is essential to demonstrate that the lack of correlation between chromosome number evolution and some of the parameters that were tested is a biological result (and therefore an important contribution because it rejects some of the current models of chromosomal evolution), and not just a lack of power of the analysis. After reading the manuscript and both reviews, I think that a stronger case needs to be made before the manuscript is suitable for publication in Proceedings B.

In particular:

1. Make a clear argument for the power of the analyses to address the questions at hand
For instance, "test whether the mean genome size for an order predicted the mean rate estimate for chromosome gains, losses, and polyploidy" seems like an extremely underpowered analysis, since you only have estimates of genome size for 4 orders. Would it not make more sense to look at correlations between chromosome numbers and genome sizes within each order? Can one really infer much from comparing 4 orders?

2. Make the analyses clearer in the manuscript

Reviewer #1 mentioned that it is not always easy for a non-expert to follow the complicated analyses, and to have a sense of how reliable they are. Both reviewer make several suggestions that would improve the clarity of the manuscript.

3. Comparisons to published data

As pointed out by reviewer #2, clearer comparisons should also be made between what is found here and what is known in the different groups. For instance, the models used here support polyploidy in every order - is this a reasonable assumption given what is known in these taxa?

Reviewer(s)' Comments to Author:

Referee: 1

Comments to the Author(s)

Please see the attached file.

I noticed that the title is misleading and somewhat strange. I tried to look for "idiosyncratic" or synonyms in the main text but failed to identify anything corresponding with the title. This is yet another attempt to sell the data using a sexy and "cool" title. The title should be modified to better reflect the actual results. Alternatively, include some summary paragraph in Discussion justifying the current title.

Referee: 2

Comments to the Author(s)

I enjoyed reading the present manuscript and I consider it a valuable contribution to our understanding of the role of genome organisation in evolution. The study is clearly a follow-up to the report published by Blackmon, Ross, and Bachtrog (2016). Therefore, the results should be compared in more detail to earlier conclusions in the discussion. Moreover, I came across several issues which need to be corrected or clarified.

line 80: "compliment" - should be "complement"?

lines 121-123: The source of sequences is not clear. Although processed alignments are available at <https://github.com/Tsylvester8/Polyneoptera>, accession numbers of original sequences of all species should be listed in Table S1. Or at least refer to the github repository.

lines 126-127 and 232: should not it be "species-level" and "genus-level" in all cases?

lines 143-144: How exactly was the inflexion point determined? I would say it is around the score of 3500.

lines 226-228 and 326-330: The authors state that 3/7 of the genera with both XY and multi-XY have a higher mean autosome number in multi-XY species, i.e. mean autosome number is increased, and this is interpreted as evidence for fissions. However, the authors reason that "if transitions from XY to multi-XY are generated by the fission of a sex chromosome we would expect no difference in the number of autosomes" for multi-XY species. This should be elaborated both in the results and discussion. I guess that the authors consider fissions responsible for increase of autosome number and therefore also for transition to multiple sex chromosome systems, although it does not fit the predicted pattern. But this is guilt by association. It was shown that autosomes and sex chromosomes evolve under different selection regimes, e.g. Ohno's law postulates that synteny of genes from the mammalian X chromosome is evolutionary more conserved compared to autosomes.

line 270: "0.063 and a polyploidy rate" to "0.063, and a polyploidy rate"

lines 311-312: "Our analyses have revealed that this clade originated from an XO ancestor." Support for a common origin, i.e. homeology, of XO sex chromosome system in Polyneoptera should be discussed (see DOI:10.1186/s12915-019-0721-x).

lines 319-320: Add references to the statement: "The transition between SCSs from XO to XY can occur through X chromosome autosome fusions or by sex chromosome turnover with fixation of the ancestral X as a new autosome."

lines 387-390: "... our analysis illustrates that rates of chromosome increase and decrease are equal in sexual and asexual Phasmatodea (Fig. 4). We interpret this as evidence that the constraints on chromosome number change via fusions, fissions, and aneuploidy are largely similar in these two groups." How could the authors determine whether the increase in chromosome number is caused by fissions or aneuploidy? Both result in increase in the haploid number by one. Could it be that there are similar rates of fissions and aneuploidies in sexuals and

asexuals, respectively? Is there any hard evidence supporting increase in chromosome number via aneuploidy in Polyneoptera?

One of interesting results of present study is the role of presumably rare polyploidy in karyotype evolution in Polyneoptera. The more complex model including polyploidy was supported by likelihood ratio tests (line 264) and relatively high polyploidy rates were estimated for Mantodea, Phasmatodea, and Orthoptera. Do available genome sizes support polyploidy inferred from chromosome numbers? As it is not always the case in insects.

Page 13, Table 1.: "A - symbol indicates a distribution of chromosome number that is uninformative or does not support a given mechanism." I cannot see any As in the table.

Author's Response to Decision Letter for (RSPB-2019-2544.R0)

See Appendix B.

RSPB-2020-1388.R0

Review form: Reviewer 2

Recommendation

Accept as is

Scientific importance: Is the manuscript an original and important contribution to its field?

Excellent

General interest: Is the paper of sufficient general interest?

Excellent

Quality of the paper: Is the overall quality of the paper suitable?

Excellent

Is the length of the paper justified?

Yes

Should the paper be seen by a specialist statistical reviewer?

No

Do you have any concerns about statistical analyses in this paper? If so, please specify them explicitly in your report.

No

It is a condition of publication that authors make their supporting data, code and materials available - either as supplementary material or hosted in an external repository. Please rate, if applicable, the supporting data on the following criteria.

Is it accessible?

Yes

Is it clear?

Yes

Is it adequate?

Yes

Do you have any ethical concerns with this paper?

No

Comments to the Author

The revised and resubmitted version of the manuscript has been much improved. Responses to my comment were satisfactory and I came across only few typos in the Supplementary materials: Suppl. Materials Line 10 - introduce the "SCS" abbreviation; Line 58 - "in Misof, Liu (13)" should probably read "in Misof et al. (13)"; Line 83 - "25 µml Propidium" should be "25 µmol Propidium?"; Line 96 - "anagentic models" should read "anagenetic models".

Decision letter (RSPB-2020-1388.R0)

17-Aug-2020

Dear Mr Sylvester

I am pleased to inform you that your manuscript RSPB-2020-1388 entitled "Idiosyncratic patterns of chromosome evolution are the rule not the exception" has been accepted for publication in Proceedings B. Congratulations!!

The referee has recommended publication, but the reviewer and Associated Editor have also suggest some minor revisions to your manuscript. Therefore, I invite you to respond to the comments and revise your manuscript. Because the schedule for publication is very tight, it is a condition of publication that you submit the revised version of your manuscript within 7 days. If you do not think you will be able to meet this date please let us know.

1) A text file of the manuscript (doc, txt, rtf or tex), including the references, tables (including captions) and figure captions. Please remove any tracked changes from the text before submission. PDF files are not an accepted format for the "Main Document".

2) A separate electronic file of each figure (tiff, EPS or print-quality PDF preferred). The format should be produced directly from original creation package, or original software format. PowerPoint files are not accepted.

3) Electronic supplementary material: this should be contained in a separate file and where possible, all ESM should be combined into a single file. All supplementary materials accompanying an accepted article will be treated as in their final form. They will be published alongside the paper on the journal website and posted on the online figshare repository. Files on figshare will be made available approximately one week before the accompanying article so that the supplementary material can be attributed a unique DOI.

Sincerely,

Dr John Hutchinson, Editor

Associate Editor

Board Member

Comments to Author:

Reviewer 2 is happy with the manuscript and only suggested very minor corrections. I have two small comments:

- I find it a little concerning that including polyploidy in the model for every clade seems to lead to the inference of unrealistic ancestral chromosome numbers in Mantodea, Orthoptera, and Phasmatodea. Do conclusions hold when polyploidy is not included for all orders?

- I agree with reviewer 1 that the title of the paper could be improved, although the short explanation of what the authors mean by idiosyncratic patterns of chromosome evolution does help a bit (but "unique" or "lineage-specific" would perhaps be clearer for many readers). More importantly, I do not feel that you can infer a "rule" from the 4 clades shown in figure 2. At the very least you should add "in Polyneoptera insects" or something like it to the title.

Reviewer(s)' Comments to Author:

Referee: 2

Comments to the Author(s).

The revised and resubmitted version of the manuscript has been much improved. Responses to my comment were satisfactory and I came across only few typos in the Supplementary materials: Suppl. Materials Line 10 - introduce the "SCS" abbreviation; Line 58 - "in Misof, Liu (13)" should probably read "in Misof et al. (13)"; Line 83 - "25 µml Propidium" should be "25 µmol Propidium?"; Line 96 - "anagentic models" should read "anagenetic models".

Decision letter (RSPB-2020-1388.R1)

03-Sep-2020

Dear Mr Sylvester

I am pleased to inform you that your manuscript entitled "Lineage-specific patterns of chromosome evolution are the rule not the exception in Polyneoptera insects" has been accepted for publication in Proceedings B. Congratulations!!

Open Access

Corresponding authors from member institutions (<http://royalsocietypublishing.org/site/librarians/allmembers.xhtml>) receive a 25% discount to these charges. For more information please visit <http://royalsocietypublishing.org/open-access>.

Paper charges

Sincerely,

Dr John Hutchinson

Associate Editor:

Board Member

Comments to Author:

I apologize for the delay in processing the final version of the manuscript, I was unexpectedly offline for a few days. The authors have done a great job at addressing the reviewer's concerns and my comments, and the paper looks great!

Appendix A

Phylogenetic data

Correct 18s and 28s to 18S and 28S.

169 (c) Modeling chromosome evolution

I was wondering about the mentioned three models to analyze genome size evolution. Please specify more. While I more or less understand authors' explanation how the analyses were conducted, I was a little more puzzled by the description of how genome size evolution was analyzed. In particular, I understood that you asked whether small vs big genomes are more prone to polyploidy. However, as we know, polyploidy itself increases genome size. How this was taken into account? I am pretty sure this simple fact was accounted for, I would just appreciate a little more information on these analyses.

Please make clear somewhere in the manuscript what is the difference between chromePlus and chromEvol, particularly for readers as myself – having some idea about these pipelines, but not knowing how exactly these are operating.

Please explain why models in ChromoSSE (Freyman and Hohna, 2017; Syst Biol) have not been tested/used.

Under (d), please recap for what traits ancestral state reconstructions were carried out.

Figure 2. Would not be possible to use more colors to differentiate chromosome number variation better? The variation is a 12-fold, however it is really hard to appreciate this in the figure with all colors sort of similar. Alternatively, one could put some important/extreme/interesting chromosome numbers outside the circle.

Discussion

8/314: here and throughout the ms.: „the rate of chromosome evolution“. What exactly is meant by this term? Should not this read “the rate of chromosome number evolution”? My understanding is that you did not analyze the rate of non-dysploid chromosomal rearrangements (reciprocal translocations, inversions,....).

Pg. 9: What I am missing when you are discussing aneuploidy is the frequency of intra-specific aneuploidy in the extant species. Is this reported? If yes, in what species and genera, and could the absence vs. presence of aneuploidy make your discussion less vague?

What is the rate of chromosome number evolution in million of years? Did I overlook this information? This seems to be missing in Discussion.

My major concern is directed towards staying just with the inferred chromosome numbers and trends of descending/ascending dysploidy. For an uninformed reader, it is impossible to realize the significance of your data set and analyses. For example, for chromosome number evolution, the performed analyses give impression that after applying the models we know what was known before. You wrote in Discussion that „We found many examples of

significantly different rates of 403 chromosome evolution among orders. Blattodea (including or excluding Isoptera) have a higher 404 rate of increases than Orthoptera (Table S3). Blattodea (excluding Isoptera) have a higher rate 405 of decrease than Isoptera and Orthoptera (Table S3). Decreases are also higher in Phasmatodea 406 than Blattodea (including Isoptera), Isoptera, and Orthoptera (Table S3). Polyploidy is higher 407 in Orthoptera than Blattodea (including Isoptera) and, Phasmatodea is higher than Blattodea 408 (including Isoptera) (Table S3).” However, it is unclear how these results compare with previously published data. Or, nothing is known about these processes? Moreover, I was wondering why some empirical data are not discussed in Results and Discussion. If you are inferring ancestral chromosome numbers, how these inferred numbers fit, for example, the most common chromosome number(s) in a given clade? What I mean here is that you do not confront your findings with previously published data/hypotheses. All this give impression that after more than 100 years of comparative (molecular) cytogenetics nothing was done in Polyneoptera. Is this really the case?

I propose to discuss if any genomes from the analyzed orders were sequenced, and if yes, what can we learn from genomic data regarding disrupted collinearity due to chromosomal rearrangements, some being dysploidal.

Appendix B

Dear Dr. Hutchinson

We would like to thank the editor and the reviewers for the positive feedback and the many useful comments on our manuscript (id: RSPB-2019-2544) " Idiosyncratic patterns of chromosome evolution are the rule not the exception". We have taken the advice of the reviewers to revise the manuscript. We have made many changes to increase the clarity of the results that were in the first version of the manuscript. We have also added three coauthors to the manuscript allowing us to include additional genome size data for 60 species that was not available in the first version of the manuscript. As one final addition to the manuscript we have deployed an interactive web database of Polyneoptera karyotype data (www.karyotype.org). This website will allow readers to explore the data used in this paper via interactive plotting or downloads for offline analyses. To remain within the page requirements of the Journal we have also moved some sections to the supplement (detailed phylogenetic inference methods, and discussions of ancestral chromosome number, etc.). We believe these changes have significantly improved the manuscript. We appreciate the careful and detailed comments that we have received and have responded to each of the reviewers' comments. Our replies are given in *red italic following each comment*. Line numbers referencing changes correspond to the new version of the manuscript without tracked changes displayed.

With kind regards,

Heath Blackmon

Editors comments:

However, it is essential to demonstrate that the lack of correlation between chromosome number evolution and some of the parameters that were tested is a biological result (and therefore an important contribution because it rejects some of the current models of chromosomal evolution), and not just a lack of power of the analysis. After reading the manuscript and both reviews, I think that a stronger case needs to be made before the manuscript is suitable for publication in Proceedings B.

Make a clear argument for the power of the analyses to address the questions at hand. For instance, "test whether the mean genome size for an order predicted the mean rate estimate for chromosome gains, losses, and polyploidy" seems like an extremely underpowered analysis, since you only have estimates of genome size for 4 orders. Would it not make more sense to look at correlations between chromosome numbers and genome sizes within each order? Can one really infer much from comparing 4 orders?

We appreciate this comment. We agree with the editor that the analysis of correlation between rates of evolution and genome size had low power. Based on this comment and one from reviewer 2 we chose to add three collaborators that were able to provide genome size estimates for an additional 60 Polyneoptera species. This additional data has been used in conjunction with species specific estimates of the rate of chromosome number evolution and chromosome number to explore the impact of genome size in a more statistically powerful context.

New data/analyses are described in:

Supplementary methods lines: 78-85
Methods lines 93-95; 141-154
Results lines 215-225
Discussion lines 316-339

Supplementary figure S3 included here:

Figure S3 Impacts of genome size. Blue lines indicate regression line without phylogenetic correction and red dashed line indicates regression line with phylogenetic correction. P-values are printed in the same color in each plot. Circles are taxa that are present in the phylogeny and triangles are taxa that are not present in the phylogeny. A) Haploid chromosome number and genome size B) Mean tip rate and genome size C) Absolute tip rate and genome size.

Make the analyses clearer in the manuscript. Reviewer #1 mentioned that it is not always easy for a non-expert to follow the complicated analyses, and to have a sense of how reliable they are. Both reviewers make several suggestions that would improve the clarity of the manuscript.

We agree parts of the text could be clearer. We have made numerous changes to clarify the text these are listed in response to individual comments below.

Comparisons to published data. As pointed out by reviewer #2, clearer comparisons should also be made between what is found here and what is known in the different groups. For instance, the models used here support polyploidy in every order - is this a reasonable assumption given what is known in these taxa?

This issue is addressed in numerous places in our response to reviewer two. In particular we have added a section to the discussion titled Reconciling a century of chromosome research on lines 341-374. In this section we compare and contrast our results to previous conclusions that had been drawn in the absence of a statistical model of karyotype and chromosome number evolution.

Referee: 1

R1.1: *I noticed that the title is misleading and somewhat strange. I tried to look for "idiosyncratic" or synonyms in the main text but failed to identify anything corresponding with the title. This is yet another attempt to sell the data using a sexy and "cool" title. The title should be modified to better reflect the actual results. Alternatively, include some summary paragraph in Discussion justifying the current title.*

We thank the reviewer for noting this. We have strengthened the connection between the title and the two possibilities we sought to test in this study 1) that phylogenetic history can explain difference in chromosome number variation among clades and 2) that each clade has unique (idiosyncratic) rates of chromosome evolution that lead to very different distributions of chromosome number among groups. Lines 80-82; 196-197.

R1.2: *Phylogenetic data
Correct 18s and 28s to 18S and 28S.*

We have corrected all the occurrences of 18s and 28s into 18S and 28S

R1.3: *169 (c) Modeling chromosome evolution
I was wondering about the mentioned three models to analyze genome size evolution. Please specify more. While I more or less understand authors' explanation how the analyses were conducted, I was a little more puzzled by the description of how genome size evolution was analyzed. In particular, I understood that you asked whether small vs big genomes are more prone to polyploidy. However, as we know, polyploidy itself increases genome size. How this was taken into account? I am pretty sure this simple fact was accounted for, I would just appreciate a little more information on these analyses.*

We agree that recent polyploidy would likely lead to increased genome size and that determining the direction of causality could be difficult. In fact, we hypothesized that recent genome duplication might drive a significant correlation between inferred rates of chromosome number evolution and genome size. However, our expanded analyses (previously 4 data points now 20 data points) show no correlation. We have changed text in lines 134-138 to make it clearer that the analyses involving genome size are testing the hypothesis that species with larger genomes (more repetitive content) would have higher rates of chromosome number change. Analyses of genome size have also been expanded to include relationships with chromosome number. These analyses are now reported in lines 141-145 and discussed in lines 323-332.

R1.4: *Please make clear somewhere in the manuscript what is the difference between chromePlus and chromEvol, particularly for readers as myself – having some idea about these pipelines, but not knowing how exactly these are operating.*

We have clarified this point in supplementary methods lines 88-98.

R1.5: *Please explain why models in ChromoSSE (Freyman and Hohna, 2017; Syst Biol) have not been tested/used.*

The focus of this study was in comparing rates across orders and within orders with different reproductive modes rather than implications for diversification. To answer these questions ChromPlus is the best option. Furthermore, Figure 6 from Freyman and Hohna suggests that a Non-SSE model (like the one we have applied) performs almost identically to a cladogenetic/anagenetic SSE model (like ChromoSSE) with regard to ancestral state reconstructions (Freyman and Höhna, 2017). Additionally, in the original publication of ChromPlus (Blackmon et al., 2019) figure 2c shows that rate estimates under SSE and nonSSE models are largely the same. Finally, it is unclear from current literature how ChromoSSE diversification estimates are impacted by the very low level of sampling that our order level analyses represent. However, we are concerned that the ChromoSSE model may behave poorly without accounting for the proportion of extant taxa that are being sampled (Davis et al., 2013). This is particularly challenging in light of uncertainty in the number of extant taxa in many insect lineages. These issues are now discussed in supplementary materials lines 88 -105. We moved this section to the supplementary materials in order to adhere with the page limit of the journal.

R1.6: *Under (d), please recap for what traits ancestral state reconstructions were carried out.*

This information has been added at lines 156-157

R1.7: *Figure 2. Would not be possible to use more colors to differentiate chromosome number variation better? The variation is a 12-fold, however it is really hard to appreciate this in the figure with all colors sort of similar. Alternatively, one could put some important/extreme/interesting chromosome numbers outside the circle.*

We agree this was difficult to interpret we have chosen to combine the former figure one and two into figure 1. We now indicate chromosome number as a vertical bar that comes out from the phylogeny making it much easier to see and appreciate the variation in chromosome number across the studied taxa.

Figure 1

R1.8: *Discussion*

8/314: here and throughout the ms.: „the rate of chromosome evolution“. What exactly is meant by this term? Should not this read “the rate of chromosome number evolution”? My understanding is that you did not analyze the rate of non-dysploid chromosomal rearrangements (reciprocal translocations, inversions,...).

We have changed “the rate of chromosome evolution” to “the rate of chromosome number evolution” throughout the manuscript.

R1.9: *Pg. 9: What I am missing when you are discussing aneuploidy is the frequency of intra-specific aneuploidy in the extant species. Is this reported? If yes, in what species and genera, and could the absence vs. presence of aneuploidy make your discussion less vague?*

We have reworked our discussion of aneuploidy and fissions lines 275-282

R1.10: *What is the rate of chromosome number evolution in million of years? Did I overlook this information? This seems to be missing in Discussion.*

The rates reported are rate parameters for the Markov models described in lines 121-126. These represent rate parameters for exponential distributions that describe the expected waiting time for transitions in the states of the Markov model. We have made this clearer in lines 127-130.

R1.11: *My major concern is directed towards staying just with the inferred chromosome numbers and trends of descending/ascending dysploidy. For an uninformed reader, it is impossible to realize the significance of your data set and analyses. For example, for chromosome number evolution, the performed analyses give impression that after applying the models we know what was knew before. You wrote in Discussion that (lines 403-408) “We found many examples of significantly different rates of chromosome evolution among orders. Blattodea (including or excluding Isoptera) have a higher rate of increases than Orthoptera (Table S3). Blattodea (excluding Isoptera) have a higher rate of decrease than Isoptera and Orthoptera (Table S3). Decreases are also higher in Phasmatodea than Blattodea (including Isoptera), Isoptera, and Orthoptera (Table S3). Polyploidy is higher in Orthoptera than Blattodea (including Isoptera) and, Phasmatodea is higher than Blattodea (including Isoptera) (Table S3).” However, it is unclear how these results compare with previously published data. Or, nothing is known about these processes? Moreover, I was wondering why some empirical data are not discussed in Results and Discussion. If you are inferring ancestral chromosome numbers, how these inferred numbers fit, for example, the most common chromosome number(s) in a given clade? What I mean here is that you do not confront your findings with previously published data/hypotheses. All this give impression that after more than 100 years of comparative (molecular) cytogenetics nothing was done in Polyneoptera. Is this really the case?*

We agree that a better connection to previous work would improve the manuscript. We have added a separate section in the discussion section titled “Reconciling a century of chromosome research” and added further information in lines 341-374. In this section we contrast the results from our model-based analysis to previous ideas that were based on either the distribution of traits in extant taxa or on less mechanistic models.

We have also added the most common chromosome number reported for each of the analyzed orders in supplementary materials lines 130-145. We moved this section to supplementary materials in order to adhere to the page limit of the Journal.

R1.12: *I propose to discuss if any genomes from the analyzed orders were sequenced, and if yes, what can we learn from genomic data regarding disrupted collinearity due to chromosomal rearrangements, some being dysploidal.*

Unfortunately, the low number and fragmentation of available genomes in these groups precludes robust comparative genomic analyses on the scale that is possible with chromosome number. However, we mention in lines 377-381 some taxa which based on our analyses may be particularly informative targets for future genome sequencing.

Referee: 2

R2.1: *I enjoyed reading the present manuscript and I consider it a valuable contribution to our understanding of the role of genome organisation in evolution. The study is clearly a follow-up to the report published by Blackmon, Ross, and Bachtrog (2016). Therefore, the results should be compared in more detail to earlier conclusions in the discussion. Moreover, I came across several issues which need to be corrected or clarified.*

Based on this comment and one from reviewer one we have added a section in the discussion that compares our results with previous hypotheses and highlights some of the insights that were not possible prior to the application of our model-based approach lines 341-374.

R2.2: *line 80: “compliment” - should be “complement”?*

Corrected

R2.3: *lines 121-123: The source of sequences is not clear. Although processed alignments are available at <https://github.com/Tsylvester8/Polyneoptera>, accession numbers of original sequences of all species should be listed in Table S1. Or at least refer to the github repository.*

We have included a table S3 that includes all accession numbers of sequences used.

R2.4: *lines 126-127 and 232: should not it be “species-level” and “genus-level” in all cases?*

Yes, corrected

R2.5: lines 143-144: *How exactly was the inflexion point determined? I would say it is around the score of 3500.*

The inflexion point was chosen by visual examination. We made this choice because there is a plateau at 4870 that is shared for several species. After this point each additional species has an increasing taxonomic instability index. This has been clarified in the supplementary materials line 46-48. We moved this section to supplementary materials in order to adhere to the page limit of the Journal.

R2.6: lines 226-228 and 326-330: *The authors state that 3/7 of the genera with both XY and multi-XY have a higher mean autosome number in multi-XY species, i.e. mean autosome number is increased, and this is interpreted as evidence for fissions. However, the authors reason that “if transitions from XY to multi-XY are generated by the fission of a sex chromosome we would expect no difference in the number of autosomes” for multi-XY species. This should be elaborated both in the results and discussion. I guess that the authors consider fissions responsible for increase of autosome number and therefore also for transition to multiple sex chromosome systems, although it does not fit the predicted pattern. But this is guilty by association. It was shown that autosomes and sex chromosomes evolve under different selection regimes, e.g. Ohno's law postulates that synteny of genes from the mammalian X chromosome is evolutionary more conserved compared to autosomes.*

We thank the reviewer for pointing this out. We have changed our table so that all comparison between species in a genus with XO and XY as well as comparisons between species with XO and multi-XY or XY and multi-XY use mean haploid autosome number. In all cases change in the sex chromosome system by fusions of an autosome and a sex chromosome should result a reduction of autosome number. In contrast, if multi-XY systems are generated by fissioning of an existing X or Y chromosomes, autosome number should remain unchanged. Based on this when we compare XY and XO lower autosome number in the XY is evidence for fusions. In the case of multi-XY and XO or XY we interpret lower autosome number in multi-XY as evidence for fusions and an equal or higher number of autosomes as evidence for fissions. Lines 172-179 and Table 1

R2.7: line 270: *“0.063 and a polyploidy rate” to “0.063, and a polyploidy rate”*

Corrected

R2.8: lines 311-312: *“Our analyses have revealed that this clade originated from an XO ancestor.” Support for a common origin, i.e. homeology, of XO sex chromosome system in Polyneoptera should be discussed (see DOI:10.1186/s12915-019-0721-x).*

We have added a citation to the suggested paper and discuss the support for a common origin of the XO sex chromosome system in Polyneoptera more fully lines 344-346.

R2.9: lines 319-320: Add references to the statement: “The transition between SCSs from XO to XY can occur through X chromosome autosome fusions or by sex chromosome turnover with fixation of the ancestral X as a new autosome.”

Added citations to the above statement at line 240-241.

R2.10: lines 387-390: “... our analysis illustrates that rates of chromosome increase and decrease are equal in sexual and asexual Phasmatodea (Fig. 4). We interpret this as evidence that the constraints on chromosome number change via fusions, fissions, and aneuploidy are largely similar in these two groups.” How could the authors determine whether the increase in chromosome number is caused by fissions or aneuploidy? Both result in increase in the haploid number by one. Could it be that there are similar rates of fissions and aneuploidies in sexuals and asexuals, respectively? Is there any hard evidence supporting increase in chromosome number via aneuploidy in Polyneoptera?

We thank the reviewer for pointing this out. In our analysis the fission rate estimated in the model could represent either fission or aneuploidy. Ks plots across insects which suggest rare but widespread support for at least partial genome duplication (might be explained by aneuploidy events). We added this information at lines 275-280 to improve our discussion on aneuploidy.

R2.11: One of interesting results of present study is the role of presumably rare polyploidy in karyotype evolution in Polyneoptera. The more complex model including polyploidy was supported by likelihood ratio tests (line 264) and relatively high polyploidy rates were estimated for Mantodea, Phasmatodea, and Orthoptera. Do available genome sizes support polyploidy inferred from chromosome numbers? As it is not always the case in insects.

We thank the reviewer for pointing this out. Ks plots provide perhaps the best evidence for polyploidy in insects and thus we have added some text discussing this in lines 275-280. Genome size differences for polyploids often decays fairly rapidly and the lack of correlation between genome size and chromosome number implies that there are likely few very recent whole genome duplications represented in our data. We have added a discussion of these points in lines 322-336.

R2.12: Page 13, Table 1.: “A - symbol indicates a distribution of chromosome number that is uninformative or does not support a given mechanism.” I cannot see any As in the table.

We apologize for this confusion the symbol “ – ” was meant to indicate clades that are uninformative. We have changed the description to make the table more easily interpreted.

Table 1

- BLACKMON, H., JUSTISON, J., MAYROSE, I. & GOLDBERG, E. E. 2019. Meiotic drive shapes rates of karyotype evolution in mammals. *Evolution*, 73, 511-523.
- DAVIS, M. P., MIDFORD, P. E. & MADDISON, W. 2013. Exploring power and parameter estimation of the BiSSE method for analyzing species diversification. *BMC evolutionary biology*, 13, 38.
- FREYMAN, W. A. & HÖHNA, S. 2017. Cladogenetic and anagenetic models of chromosome number evolution: A Bayesian model averaging approach. *Systematic biology*, 67, 195-215.